# A Singular Stochastic Control Approach for Optimal Pairs Trading with Proportional Transaction Costs

Haipeng Xing

Department of Applied Mathematics and Statistics, Stony Brook University, Stony Brook, NY 11794, USA; haipeng.xing@stonybrook.edu; Tel.: +1-631-632-1892

**Abstract:** Optimal trading strategies for pairs trading have been studied by models that try to find either optimal shares of stocks by assuming no transaction costs or optimal timing of trading fixed numbers of shares of stocks with transaction costs. To find optimal strategies that determine optimally both trade times and number of shares in a pairs trading process, we use a singular stochastic control approach to study an optimal pairs trading problem with proportional transaction costs. Assuming a cointegrated relationship for a pair of stock log-prices, we consider a portfolio optimization problem that involves dynamic trading strategies with proportional transaction costs. We show that the value function of the control problem is the unique viscosity solution of a nonlinear quasi-variational inequality, which is equivalent to a free boundary problem for the singular stochastic control value function. We then develop a discrete time dynamic programming algorithm to compute the transaction regions, and show the convergence of the discretization scheme. We illustrate our approach with numerical examples and discuss the impact of different parameters on transaction regions. We study the out-of-sample performance in an empirical study that consists of six pairs of U.S. stocks selected from different industry sectors, and demonstrate the efficiency of the optimal strategy.

**Keywords:** free-boundary problem; pairs trading; stochastic control; trading strategies; transaction costs; transaction regions

## 1. Introduction

Pairs trading is one of proprietary statistical arbitrage tools used by many hedge funds and investment banks. It is a short-term trading strategy that first identifies two stocks whose prices are associated in a long-run equilibrium and then trades on temporary deviations of stock prices from the equilibrium. Though pairs trading is a simple market neutral strategy, it has been used and discussed extensively by industrial practitioners in the last several decades; see detailed discussion in Vidyamurthy (2004), Whistler (2004), Ehrman (2006), Lai and Xing (2008), and references therein.

Besides its wide practice in financial industry, pairs trading also draws much attention from academic researchers. For instance, Gatev et al. (2006) examined the risk and returns of pairs trading using daily data collected from the U.S. equity market and concluded that the strategy in general produces profit higher than transaction costs. To investigate the pairs trading strategy analytically, Elliott et al. (2005) modeled the spread of returns as a mean-reverting process and proposed a trading strategy based on the model. This motivates subsequent researchers to formulate pairs trading rules as stochastic control problems for an Ornstein–Uhlenbeck (OU) process and a correlated stock price process. In particular, Mudchanatongsuk et al. (2008) assumed the log-relationship between a pair of stock prices follows a mean-reverting process, and considered a self-financing portfolio strategy that only allows positions that were long in one stock and short in the other with equal dollar amounts. They then formulated a portfolio optimization based stochastic control problem and obtained the optimal solution to this control problem in closed form via the corresponding Hamilton–Jacobi–Bellman (HJB) equation. Relaxing the

equal dollar constraint, Tourin and Yan (2013) extended Mudchanatongsuk et al. (2008)'s approach and study pairs trading strategies with arbitrary amounts in each stock without any transaction costs.

Instead of deriving optimal weights of stocks in pairs trading, another line of study on pairs trading strategies fixes the number of traded shares for each stock during the entire trading process and considers only the optimal timing of trades in the presence of fixed or proportional transaction costs. Specifically, Leung and Li (2015) studies the optimal timing to open or close the position subject to fixed transaction costs and the effect of stop-loss level under the OU process by constructing the value function directly. Zhang and Zhang (2008), Song and Zhang (2013), and Ngo and Pham (2016) studied the optimal pairs trading rule that is based on optimal switching among two (buy and sell) or three (buy, sell, and flat) regimes with a fixed commission cost for each transaction, and solve the problem by finding viscosity solutions to the associated HJB equations (quasi-variational inequalities). Lei and Xu (2015) studied the optimal pairs trading rule of entering and exiting the asset market in finite horizon with proportional transaction cost for two convergent assets. Note that, although transaction costs are considered in these strategies, since the number of traded shares of stocks are fixed during the entire trading period, these strategies are still far from traders' practical experience in reality.

The above study on optimal pairs trading focuses either on optimal trading shares without transaction costs or optimal trading times with fixed trading shares in the presence of transaction costs. To relax the assumption of fixed trading shares in the latter study, this paper uses a singular stochastic control approach to study the joint effect of optimal trading shares and optimal trading times in pairs trading process with proportional transaction costs. For convenience, we assume the same diffusion and Urnstein–Uhlenbeck processes for one stock and its spread with the other stock as those in Mudchanatongsuk et al. (2008). However, different from Mudchanatongsuk et al. (2008) who used a trading rule which requires to short one stock and long the other in equal dollar amounts, we consider a delta-neutral rule under which the ratio of traded shares for two stocks is fixed and this fixed ratio is determined by the cointegration relationship of two stocks. Hence, when the number of shares of one stock is determined, based on the rule of delta neutral, the number of shares for the other stock is also determined. Besides the weight of shares need to be optimally chosen, we also assume a proportional transaction cost for each trade and hence the optimal times of trading also needs to be decided.

With the above assumptions, we solve the optimal pair trading problem by the singular stochastic control approach in Davis et al. (1993). As the overall transaction cost based on the above assumption depends on both trading times and the numbers of shares in each trade, we compute the terminal utility of wealth over a fixed horizon and formulate the problem of choosing optimal trading times and the number of shares as a singular stochastic control problem. We derive the Hamilton–Jacobi–Bellman equations for this problem, and show that the value function of the problem is the unique viscosity solution of a quasi-variational inequality. We further argue that the quasi-variational inequality is equivalent to a free boundary problem so that the state space consisting of one stock price and its spread with the other stock can be naturally divided into three transaction regions: long the first stock and short the second, short the first and long the second, and no transaction. The implied transaction regions can help us determine not only optimal times of each transaction, but also the optimal number of shares in each transaction. To compute the boundaries of these transaction regions, we develop a numerical algorithm that is based on discrete time dynamic programming to solve the equation for the negative exponential utility function, and show that the numerical solution converges to the unique continuous-time solution of the problem.

To demonstrate the advantage of joint consideration of optimal shares and optimal trading times in pair trading, we carry out both simulation and empirical studies. Specifically, we study the time-varying transaction regions (or trading boundaries) for a specific set of model parameters, and investigate the impact of variations of model parameters on

transaction regions and performance of the optimal strategy. For comparison purposes, we also consider a benchmark strategy based on the deviation of the spread from its long-term mean and is popular among practitioners. In both simulation studies and real data analysis, we show that the optimal trading strategy performs better than the benchmark strategy.

The rest of the paper is organized as follows. Section 2 first formulates the model and then derive the Hamilton–Jacobi–Bellman equations associated with the singular stochastic control problems. It shows the existence and uniqueness of the viscosity solution for the variational inequalities, which are equivalent to the portfolio optimization problem, and reduces the problem into a free boundary problem. Section 2 also considers the optimal trading problem with exponential utility functions. In Section 3, we discretize the free boundary problem and propose a discrete time dynamic programming algorithm. We also demonstrate that the solution of the discretized problem converges to the viscosity solution of the variational inequalities. Sections 4 and 5 provide simulation and empirical studies of the model and the optimal trading strategy, and compare its performance with a benchmark trading strategy. Some concluding remarks are given in Section 6.

## 2. A Pairs Trading Problem with Proportional Transaction Costs

### 2.1. Model Specification

Consider a pair of two stocks $P$ and $Q$, and let $p(t)$ and $q(t)$ denote their prices at time $t$, respectively. We assume that the price of stock $P$ follows a geometric Brownian motion,

$$dp(t) = \mu p(t)dt + \sigma p(t)dB(t), \tag{1}$$

where $\mu$ and $\sigma$ are the drift and the volatility of stock $P$, and $B(t)$ is a standard Brownian motion defined on a filtered probability space and specified later. Denote $x(t)$ the difference of the logarithms of the two stock prices, i.e.,

$$x(t) = \log q(t) - \log p(t) = \log(q(t)/p(t)). \tag{2}$$

We assume that the spread follows an Ornstein–Uhlenbeck process

$$dx(t) = \kappa(\theta - x(t))dt + \nu dW(t), \tag{3}$$

where $\kappa > 0$ is the speed of mean reversion, and $\theta$ is the long-term equilibrium level to which the spread reverts. We assume that $(B(t), W(t))$ is a two-dimensional Brownian motion defined on a filtered probability space $(\Omega, \mathcal{F}_t, \mathbb{P})$, and the instantaneous correlation coefficient between $B(t)$ and $W(t)$ is $\rho$, i.e.,

$$E[dW(t)dB(t)] = \rho dt. \tag{4}$$

The above assumptions are same as those in Mudchanatongsuk et al. (2008). With these assumptions, we can express the dynamics of $q(t)$ as

$$dq(t) = \left[\mu + \kappa(\theta - x(t)) + \frac{1}{2}\nu^2 + \rho\sigma\nu\right]q(t)dt + \sigma q(t)dB(t) + \nu q(t)dW(t). \tag{5}$$

In the presence of proportional transaction costs, the investor pays $0 < \zeta_p, \zeta_q < 1$ and $0 < \eta_p, \eta_q < 1$ of the dollar value transacted on purchase and sale of the underlying stocks $P$ and $Q$. Denote $L_p(t)$ and $M_p(t)$ two nondecreasing and non-anticipating processes and represent the cumulative number of shares of stock $P$ bought or sold, respectively, within the time interval $[0, t]$, $0 \leq t \leq T$. Let $y_p(t)$ be the number of shares held in stock $P$, i.e., $y_p(t) = L_p(t) - M_p(t)$, and similarly, we define $L_q(t)$, $M_q(t)$, and $y_q(t) = L_q(t) - M_q(t)$ for stock $Q$. Denote $g(t)$ the dollar value of the investment in bond which pays a fixed risk-free rate of $r$. Then, the investor's position in two stocks and the bond is driven by

$$dy_p(t) = dL_p(t) - dM_p(t), \qquad dy_q(t) = dL_q(t) - dM_q(t) \tag{6}$$

and

$$dg(t) = rg(t)dt + b_p p(t)dM_p(t) - a_q q(t)dL_q(t) + b_q q(t)dM_q(t) - a_p p(t)dL_p(t), \quad (7)$$

where $a_i = 1 + \zeta_i$ and $b_i = 1 - \eta_i$ for $i = p, q$.

We then need to choose a rule to determine the number of shares of stocks $P$ and $Q$ bought or sold at time $t$. Note that, Mudchanatongsuk et al. (2008) assumed no transaction cost and considered the strategy that always shorts one stock and longs the other in equal dollar amount, i.e., $p(t)dL_p(t) + q(t)dM_q(t) = 0$ or $p(t)dM_p(t) + q(t)dL_q(t) = 0$ at time $t$. Lei and Xu (2015) and Ngo and Pham (2016) considered a delta-neutral strategy that always long one share of a stock and short one share of the other stock, i.e., $dy_p(t) = -dy_q(t) = 1$ or $dy_p(t) = -dy_q(t) = -1$ at time $t$. Here, we also consider a delta-neutral strategy that requires the total of positive and negative delta of two assets is zero, hence it suggests that the number of shares of stock $P$ bought (or sold) at time $t$ are same as the number of shares of stock $Q$ sold (or bought), i.e.,

$$dL_p(t) = dM_q(t), \qquad dM_p(t) = dL_q(t). \quad (8)$$

Equation (8) implies that

$$dy_q(t) = -dy_p(t)$$

at any time $t$. Comparing to Lei and Xu (2015) and Ngo and Pham (2016), we remove the constraint $dy_p(t) = -dy_q(t) = 1$ or $-1$ and allow $y_p(t) = -y_q(t)$ to be a control variable. Using Equations (5) and (8), the dynamics of $g(t)$ in Equation (7) can be simplified as

$$dg(t) = rg(t)dt - \left(a_p - b_q e^{x(t)}\right)p(t)dL_p(t) + \left(b_p - a_q e^{x(t)}\right)p(t)dM_p(t). \quad (9)$$

The process $(L_p(t), M_p(t))$ together with our delta-neutral strategy provides us an admissible trading strategy. For convenience, we denote $\mathcal{T}(g_0)$ the set of admissible trading strategies that an investor starts at time zero with amount $g_0$ of the investment in bond and zero holdings in two stocks (i.e., $y_p(0) = y_q(0) = 0$), which indicates that the numbers of shares held in stocks $P$ and $Q$ at time $t$ are $y_p(t)$ and $-y_p(t)$, respectively. For notational convenience, we omit the subscript of $y_p(t)$ and denote $y_p(t)$ as $y(t)$ in our discussion. Then, Equations (1), (3), (6) and (9) compose the market model in the time interval $[0, T]$, which describes a stochastic process of $(p(t), x(t), y_p(t), g(t))$ in $\mathbb{R}^+ \times \mathbb{R} \times \mathbb{R} \times \mathbb{R}$.

Denote the terminal value of the pairs trading portfolio by $J(x(T), p(T), y(T))$. Note that, under our assumption, $y(T)$ indicates that the investor's positions in stocks $P$ and $Q$ are $y(T)$ and $-y(T)$, respectively, then the liquidated value of the portfolio is

$$J(p(T), x(T), y(T)) = A_+(p(T), x(T))y(T)\mathbb{1}_{\{y(T) \geq 0\}} + A_-(p(T), x(T))y(T)\mathbb{1}_{\{y(T) < 0\}}, \quad (10)$$

where

$$A_+(p, x) = (b_p - a_q e^x)p, \qquad A_-(p, x) = (a_p - b_q e^x)p.$$

Furthermore, if the investment in bond at terminal time $T$ is $g(T)$, the terminal wealth of the investor is given by $g(T) + J(p(T), x(T), y(T))$. Suppose that the investor's utility $U : \mathbb{R} \longrightarrow \mathbb{R}$ is a concave and increasing function with $U(0) = 0$. We assume that the investor's goal is to maximize the expected utility of terminal wealth under the market model (1), (3), (6) and (9),

$$V(t, p, x, y, g) = \sup_{(L_p(t), M_p(t)) \in \mathcal{T}(g_0)} E\Big\{ U(g(T) + J(p(T), x(T), y(T)))|p(t) = p,$$

$$x(t) = x, y_t = y, g(t) = g \Big\}. \quad (11)$$

Furthermore, given trading strategies $(L_p, M_p)$, the total trading cost incurred over $[t, T]$ can be expressed as

$$C(L_p, M_p; t, T) = \int_t^T e^{r(T-u)} A_-(p(u), x(u)) dL_p(u) - \int_t^T e^{r(T-u)} A_+(p(u), x(u)) dM_p(u)$$
$$- J(p(T), x(T), y(T)). \tag{12}$$

and the total profit over $[t, T]$ is $-C(L_p, M_p; t, T)$.

### 2.2. The Hamilton-Jacobi-Bellman Equations and Free Boundary Problems

We now derive the Hamilton–Jacobi–Bellman (HJB) equations, associated with the stochastic control problems, for the utility maximization problem (11). Consider a class of trading strategies such that $L_p(t)$ and $M_p(t)$ are absolutely continuous processes, given by

$$L_p(t) = \int_0^t l(u) du, \qquad M_p(t) = \int_0^t m(u) du,$$

where $l(u)$ and $m(u)$ are positive and uniformly bounded by $\xi < \infty$. Then, (1), (3), (6) and (9) provides us a system of stochastic differential equations with controlled drift, and the Bellman equation for a value function denoted by $V^\xi$ is

$$\mathcal{L}_{1,o} V^\xi + \sup_{0 \le l_t, m_t \le \xi} \left\{ \left[ \mathcal{L}_{1,b} V^\xi \right] l_t - \left[ \mathcal{L}_{1,s} V^\xi \right] m_t \right\} = 0,$$

for $(t, p, X, y, g) \in [0, T] \times \mathbb{R}^+ \times \mathbb{R} \times \mathbb{R} \times \mathbb{R}$, in which the operators $\mathcal{L}$, $\mathcal{B}$, and $\mathcal{S}$ are defined as

$$\mathcal{L}_{1,o} := \frac{\partial}{\partial t} + \kappa(\theta - x) \frac{\partial}{\partial x} + \mu p \frac{\partial}{\partial p} + rg \frac{\partial}{\partial g} + \frac{1}{2} v^2 \frac{\partial^2}{\partial x^2} + \rho v \sigma p \frac{\partial^2}{\partial p \partial x} + \frac{1}{2} \sigma^2 p^2 \frac{\partial^2}{\partial p^2},$$

$$\mathcal{L}_{1,b} := \frac{\partial}{\partial y} - (a_p - b_q e^{x(t)}) p(t) \frac{\partial}{\partial g},$$

$$\mathcal{L}_{1,s} := \frac{\partial}{\partial y} - (b_p - a_q e^{x(t)}) p(t) \frac{\partial}{\partial g}.$$

The optimal trading strategy is then determined by considering the following three possible cases:

(i)    buying stock $P$ and sell stock $Q$ at the same rate $l(t) = \xi$ (i.e., $m(t) = 0$) when

$$\mathcal{L}_{1,b} V^\xi \ge 0, \qquad \mathcal{L}_{1,s} V^\xi > 0; \tag{13}$$

(ii)    selling stock $P$ and buy stock $Q$ at rate $m(t) = \xi$ (i.e., $l(t) = 0$) when

$$\mathcal{L}_{1,b} V^\xi < 0, \qquad \mathcal{L}_{1,s} V^\xi \le 0; \tag{14}$$

(iii)    doing nothing (i.e., $l(t) = m(t) = 0$) when

$$\mathcal{L}_{1,b} V^\xi \le 0, \qquad \mathcal{L}_{1,s} V^\xi \ge 0. \tag{15}$$

Note that the case $\mathcal{L}_{1,b} V^\xi > 0$ and $\mathcal{L}_{1,s} V^\xi < 0$ can not occur, as all value functions are increasing functions of $g$.

The above argument shows that the optimization problem (11) is a free boundary problem in which the optimal trading strategy is defined by the inequalities (i), (ii), and (iii) for a given value function. Besides, the state space $[0, T] \times \mathbb{R}^+ \times \mathbb{R} \times \mathbb{R} \times \mathbb{R}$ is partitioned into *buy*, *sell*, and *no-transaction* regions for stock $P$, which are characterized by inequalities (13), (14) and (15), respectively. For sufficiently large $\xi$, the state space remains divided into a *buy region* $\mathcal{B}$, a *sell region* $\mathcal{S}$, and a *no-transaction region* $\mathcal{N}$ for stock $P$, which

are correspondingly the *sell region*, the *buy region*, and the *no transaction region* for stock $Q$ due to Equation (8). Obviously, the buy and sell regions for stock $P$ are disjoint, as it is not optimal to buy and sell the same stock at the same time. We denote the boundaries between the no-transaction region $\mathcal{N}$ and the buy and sell regions $\mathcal{B}$ and $\mathcal{S}$ as $\partial\mathcal{B}$ and $\partial\mathcal{S}$, respectively.

Let $\xi \to \infty$, the class of admissible trading strategies becomes $\mathcal{T}(g_0)$. We can guess that the state space is still divided into three regions, a region of buying $P$ and selling $Q$, a region of selling $P$ and buying $Q$, and a no-transaction region. Then, the optimal trading strategy requires an immediate move to the boundaries of buy or sell regions, if the state is in the buy region $\mathcal{B}$ or the sell region $\mathcal{S}$. Actually, we can obtain equations that each of the value functions should satisfy as follows.

(i) In region $\mathcal{B}$ of buying $P$ and selling $Q$, the value function remains constant along the path of the state, dictated by the optimal trading strategy, and therefore, for $\delta y \geq 0$

$$V(t, p, x, y, g) = V(t, p, x, y + \delta y, g - (a_p - b_q e^x)p\delta y), \tag{16}$$

where $\delta y$ is the number of shares of stock $P$ bought and stock $Q$ sold by the investor. $\delta y$ can be any positive value up to the number required to take the state to $\partial\mathcal{B}$, so letting $\delta y \to 0$ in (16) yields

$$\mathcal{L}_{1,b}V = 0. \tag{17}$$

(ii) Similarly, in region $\mathcal{S}$ of selling $P$ and buying $Q$, the value function obeys the following equation for $\delta y \geq 0$

$$V(t, p, x, y, g) = V(t, p, x, y - \delta y, g + (b_p - a_q e^x)p\delta y), \tag{18}$$

where $\delta y$ is the number of shares of stock $P$ sold and stock $Q$ bought by the investor. $\delta y$ can be any positive value up to the number required to take the state to $\partial\mathcal{S}$, so letting $\delta y \to 0$ in (18) yields

$$\mathcal{L}_{1,s}V = 0. \tag{19}$$

(iii) In the no-transaction region, the value function obeys the same set of equations obtained for the class of absolutely continuous trading strategies, and therefore the value function is given by

$$\mathcal{L}_{1,o}V = 0, \tag{20}$$

and the pair of inequalities, shown above in (15), also hold. Note that, due to the continuity of the value function, if it is known in the no-transaction region, it can be determined in both the buy and sell regions by (17) and (19), respectively.

In the buy region $\mathcal{B}$, $\mathcal{L}_{1,s}V < 0$, and, in the sell region $\mathcal{S}$, $\mathcal{L}_{1,b}V > 0$. Additionally, from the two pairs of inequalities (13) and (14), we may conjecture that $\mathcal{L}_{1,o}V$ in (20) is negative in both the buy region $\mathcal{B}$ and the sell region $\mathcal{S}$. Therefore, the above set of equations can be summarized as the following fully nonlinear partially differential equations (PDE):

$$\min\left\{-\mathcal{L}_{1,b}V, \mathcal{L}_{1,s}V, -\mathcal{L}_{1,o}V\right\} = 0 \tag{21}$$

for $(t, p, X, y, g) \in [0, T] \times \mathbb{R}^+ \times \mathbb{R} \times \mathbb{R} \times \mathbb{R}$. Note that the above discussion also yields the following free boundary problem for the singular stochastic control value function:

$$\begin{cases} \mathcal{L}_{1,b}V &= 0 & \text{in } \mathcal{B} \\ \mathcal{L}_{1,s}V &= 0 & \text{in } \mathcal{S} \\ \mathcal{L}_{1,o}V &= 0 & \text{in } \mathcal{N} \\ V(T, p, x, y, g) &= U(g + J(p, x, y)). \end{cases} \tag{22}$$

We next show that the value function given by (11) is a constrained viscosity solution of the variational inequality (21) on $[0, T] \times \mathbb{R}^+ \times \mathbb{R} \times \mathbb{R} \times \mathbb{R}$, and it is the unique bounded constrained viscosity solution of (21). The proof is given in the Appendix A.

**Theorem 1.** *The value function $V(t, p, x, y, g)$ is a constrained viscosity solution of* (21) *on* $[0, T] \times \mathbb{R}^+ \times \mathbb{R} \times \mathbb{R} \times \mathbb{R}$.

**Theorem 2.** *Let $u$ be a bounded upper semicontinuous viscosity subsolution of* (21)*, and $v$ a bounded from below lower semicontinuous viscosity supersolution of* (21)*, such that $u(T, \mathbf{x}) \leq v(T, \mathbf{x})$ for all $\mathbf{x} \in \mathbb{R}^+ \times \mathbb{R} \times \mathbb{R} \times \mathbb{R}$. Then $u \leq v$ on $[0, T] \times \mathbb{R}^+ \times \mathbb{R} \times \mathbb{R} \times \mathbb{R}$.*

*2.3. Optimal Trading with Exponential Utility Functions*

We next assume that the investor has the negative exponential utility function

$$U(z) = 1 - \exp(-\gamma z), \tag{23}$$

where $\gamma$ is the constant absolute risk aversion (CARA) parameter such that $-U''(z)/U'(z) = \gamma$. For Equation (21), this utility function can reduce much of computational effort and is easy to interpret. Note that for the utility function (23), the definition of the value function (11) can be expressed as

$$V(t, p, x, y, g) = 1 - \exp\left(-\gamma g e^{r(T-t)}\right) H(t, p, x, y), \tag{24}$$

where $H(t, p, x, y)$ is a convex nonincreasing continuous function in $y$ and defined by

$$
\begin{aligned}
H(t, p, x, y) &= \inf_{Lp(t), M_p(t) \in \mathcal{T}(g_0)} E\left\{ \exp[-\gamma J(p(T), x(T), y(T))] \big| p(t) = p, x(t) = x, y(t) = y \right\} \\
&= 1 - V(t, p, x, y, 0).
\end{aligned}
$$

Plug (24) into (21), and define the following operators for $H(t, p, x, y)$ on $[0, T] \times \mathbb{R}^+ \times \mathbb{R} \times \mathbb{R}$,

$$\mathcal{L}_{2,o}H = \frac{\partial H}{\partial t} + \kappa(\theta - x)\frac{\partial H}{\partial x} + \mu p \frac{\partial H}{\partial p} + \frac{1}{2}v^2 \frac{\partial^2 H}{\partial x^2} + \rho v \sigma p \frac{\partial^2 H}{\partial p \partial x} + \frac{1}{2}\sigma^2 p^2 \frac{\partial^2 H}{\partial p^2},$$

$$\mathcal{L}_{2,b}H = \frac{\partial H}{\partial y} + \gamma e^{r(T-t)} A_-(p, x) H,$$

$$\mathcal{L}_{2,s}H = \frac{\partial H}{\partial y} + \gamma e^{r(T-t)} A_+(p, x) H.$$

Then (21) is transformed into the following PDE for $H(t, p, x, y)$

$$\min\left\{ \mathcal{L}_{2,b}H, -\mathcal{L}_{2,s}H, \mathcal{L}_{2,o}H \right\} = 0 \tag{25}$$

with the following boundary conditions

$$H(T, p, x, y) = \exp\left\{ -\gamma J(p, x, y) \right\}.$$

Correspondingly, the free boundary problem (22) becomes

$$
\begin{cases}
\mathcal{L}_{2,o}H &= 0 & y \in [Y_b(t, p, x), Y_s(t, p, x)] \\
\mathcal{L}_{2,b}H &= 0 & y \leq Y_b(t, p, x) \\
\mathcal{L}_{2,s}H &= 0 & y \geq Y_s(t, p, x) \\
H(T, p, x, y) &= \exp\left\{ -\gamma J(p, x, y) \right\}.
\end{cases}
\tag{26}
$$

in which $Y_b(t, p, x)$ and $Y_s(t, p, x)$ are the buy and sell boundaries for stock $P$, respectively. Note that the function $H(t, p, x, y)$ is evaluated in the four-dimensional space $[0, T] \times \mathbb{R} \times \mathbb{R} \times \mathbb{R}$. Furthermore, this suggests that while $(t, u_t, w_t)$ is inside the no-transaction region, the dynamics of $h(t, u, w, y)$ are driven by two-dimensional standard Brownian motions

$\{z_t, t \geq 0\}$ and $\{w_t, t \geq 0\}$ with correlation $\rho$. In the buy and sell regions, it follows from (26) that

$$H(t, p, x, y) = \exp\{-\gamma e^{r(T-t)} A_-(p, x)[y - Y_b(t, p, x)]\} H(t, p, x, Y_b(t, p, x)), \quad y \leq Y_b(t, p, x),$$

$$H(t, p, x, y) = \exp\{-\gamma e^{r(T-t)} A_+(p, x)[y - Y_s(t, p, x)]\} H(t, p, x, Y_s(t, p, x)), \quad y \geq Y_s(t, p, x).$$

## 3. Discretization and a Numerical Algorithm

The solution of the PDE (21) or (25) can be obtained by turning the stochastic differential Equations (1), (3), (6) and (9) into Markov chains and then applying the discrete time dynamic programming algorithm. The discrete state is $\mathbb{X} = (\chi, \mathbb{p}, \mathbb{x}, \vartheta, \mathbb{g})$, whose elements denote time, price of stock $P$, spread, number of shares of stock $P$, and amount in the bank in a discrete space. The value function, denoted by $\mathbb{V}$, are given a value at the final time by using the boundary conditions for the continuous value functions over the discrete subspace $(\mathbb{p}, \mathbb{x}, \vartheta, \mathbb{g})$, and then they are estimated by proceeding backward in time by using the discrete time algorithm. As in the continuous time case, this algorithm is the same for both value functions and is derived below for a value function denoted by $\mathbb{V}^\delta(\chi, \mathbb{p}, \mathbb{x}, \vartheta, \mathbb{g})$, where $\rho$ is a discretization parameter, which depends on the discrete time interval $t_\delta$. If $t_\delta$ and the resolution of the $\vartheta$-axis $\vartheta_\delta$ are sent to zero, then the above discrete value function converges to a viscosity subsolution and a viscosity supersolution of the PDE (21). Therefore, all the discrete value functions converge to their continuous counterparts; this is due to the uniqueness of the viscosity solution.

Consider an evenly spaced partition of the time interval $[0, T]$: $\chi = \{\delta, 2\delta, \ldots, n\delta\}$, where $\delta = T/n$, and two evenly spaced partitions of the space intervals $\mathbb{z} = \{0, \pm\sqrt{\delta}, \pm 2\sqrt{\delta}, \ldots, \}$ and $\mathbb{w} = \{0, \pm\sqrt{\delta}, \pm 2\sqrt{\delta}, \ldots, \}$. The grid $\mathbb{p}$ is defined by $\mathbb{z}$ via the following transformation,

$$\mathbb{p}_i = \exp\left((\mu - \frac{1}{2}\sigma^2)T + \mathbb{z}_i \sigma\sqrt{T}\right). \tag{27}$$

Note that the SDE (3) implies that the asymptotic distribution of $X(t)$ is Normal $(\theta, \nu^2/(2\kappa))$, we define grid $\mathbb{x}$ by

$$\mathbb{x}_j = \theta + \frac{\nu}{\sqrt{2\kappa}} \mathbb{w}_j. \tag{28}$$

Denote $\chi_i = i\delta$ for $i = 1, \ldots, n - 1$. The dynamics (1) and (3) of $P(t)$ and $X(t)$ implies the following transition density for $(\mathbb{p}(\chi_i), \mathbb{x}(\chi_i))$,

$$\begin{pmatrix} \mathbb{p}(\chi_{i+1}) \\ \mathbb{x}_{\chi_{i+1}} \end{pmatrix} \Bigg| \begin{pmatrix} \mathbb{p}(\chi_i) \\ \mathbb{x}_{\chi_i} \end{pmatrix} \sim N\left(\begin{pmatrix} \log \mathbb{p}(\chi_i) + (\mu - \frac{1}{2}\sigma^2)\delta \\ (1 - \delta\kappa)\mathbb{x}(\chi_i) + \delta\kappa\theta \end{pmatrix}, \begin{pmatrix} \delta\sigma^2, \delta\rho\sigma\nu \\ \delta\rho\sigma\nu, \delta\nu^2 \end{pmatrix}\right). \tag{29}$$

We also note that the discrete time equation for the amount in the bank $\mathbb{g}(\chi)$ is

$$\mathbb{g}(\chi_{i+1}) = \mathbb{g}(\chi_i) \exp(r\delta).$$

Given the grid defined above, the discrete time dynamic programming principle is invoked, and the following discretization scheme is proposed for PDE (21):

$$\mathbb{V}^\delta(\chi_i, \mathbb{p}(\chi_i), \mathbb{x}(\chi_i), \vartheta, \mathbb{g}(\chi_i)) = \max\Big\{$$
$$\mathbb{V}^\delta(\chi_i, \mathbb{p}(\chi_i), \mathbb{x}(\chi_i), \vartheta + \xi, \mathbb{g}(\chi_i) - (a_p - b_q e^{\mathbb{x}(\chi_i)})\mathbb{p}(\chi_i)\xi),$$
$$\mathbb{V}^\delta(\chi_i, \mathbb{p}(\chi_i), \mathbb{x}(\chi_i), \vartheta - \xi, \mathbb{g}(\chi_i) + (b_p - a_q e^{\mathbb{x}(\chi_i)})\mathbb{p}(\chi_i)\xi), \tag{30}$$
$$E\{\mathbb{V}^\delta(\chi_{i+1}, \mathbb{p}(\chi_{i+1}), \mathbb{x}(\chi_{i+1}), \vartheta, \mathbb{g}(\chi_{i+1}))\}\Big\}.$$

where $\xi > 0$ is a real constant and $i = 0, \ldots, n - 1$. This scheme is based on the principle that the investor's policy is the choice of the optimum transaction. We next show that, as

the discretization parameter $\delta \to 0$, the solution $\mathbb{V}^\delta$ of (30) converges to the value function $V$, or, equivalently, to the unique constrained viscosity solution of (21).

**Theorem 3.** *The solution* $\mathbb{V}^\delta$ *of* (30) *converges locally uniformly as* $\delta \to 0$ *to the unique continuous constrained viscosity solution of* (21).

For the exponential utility function $U(z) = 1 - \exp(-\gamma z)$, the value function $V$ can be expressed as (24), and its discretization scheme is given by

$$\mathbb{V}^\delta(\chi_i, \mathbb{p}(\chi_i), \mathbb{x}(\chi_i), \vartheta, \mathbb{g}(\chi_i)) = 1 - \exp\left(-\gamma \mathbb{g}(\chi_i) e^{r(T-\chi_i)}\right) \mathbb{H}^\delta(\chi_i, \mathbb{p}(\chi_i), \mathbb{x}(\chi_i), \vartheta).$$

Then, the discretization scheme (30) can be reduced to

$$\mathbb{H}^\delta(\chi_i, \mathbb{p}(\chi_i), \mathbb{x}(\chi_i), \vartheta) = \min\Big\{ F_b(\mathbb{p}(\chi_i), \mathbb{x}(\chi_i), \xi) \cdot \mathbb{H}^\delta(\chi_i, \mathbb{p}(\chi_i), \mathbb{x}(\chi_i), \vartheta + \xi),$$
$$F_s(\mathbb{p}(\chi_i), \mathbb{x}(\chi_i), \xi) \cdot \mathbb{H}^\delta(\chi_i, \mathbb{p}(\chi_i), \mathbb{x}(\chi_i), \vartheta - \xi), \; E\big\{\mathbb{H}^\delta(\chi_{i+1}, \mathbb{p}(\chi_{i+1}), \mathbb{x}(\chi_{i+1}), \vartheta)\big\} \Big\}. \tag{31}$$

where

$$F_b(\mathbb{p}(\chi_i), \mathbb{x}(\chi_i), \xi) = \exp\big\{\gamma \xi A_-(\mathbb{p}(\chi_i), \mathbb{x}(\chi_i)) e^{r(T-\chi_i)}\big\},$$
$$F_s(\mathbb{p}(\chi_i), \mathbb{x}(\chi_i), \xi) = \exp\big\{-\gamma \xi A_+(\mathbb{p}(\chi_i), \mathbb{x}(\chi_i)) e^{r(T-\chi_i)}\big\}.$$

## 4. Simulation Studies

### 4.1. Buy and Sell Regions

We use the numerical algorithm proposed in Section 2 to studies the buy and sell boundaries of the pairs trading strategy. Our study focuses on two aspects of the problem. The first is the property of buy and sell boundaries (or no transaction regions) for a given set of model parameters, and the other is the impact of different model parameters on the shape of buy and sell boundaries. Without loss of the generality, we assume the time horizon $T = 1$ and $p(0) = 1$ in all our simulation studies.

We first consider a baseline scenario. The parameter values in the baseline scenario are $\mu = 0.2, \sigma = 0.4, \theta = 0.1, \kappa = 1, \nu = 0.15, \rho = 0.5, r = 0.01, \gamma = 5$ and $\zeta_p = \zeta_q = \xi_p = \xi_q = 0.0005$. For convenience, we label the setting of the baseline parameter values as Scenario 1 or (S1). We discretize the state space $(t, p, x, y, g)$ and use the developed Markov chain approximation to solve the discretized optimization problem. Figure 1 shows the buy and sell surfaces of (S1) at time $t = 0.05, 0.35, 0.65$, and 0.95. To better read the figure, we also show in Figures 2 and 3 the buy and sell boundaries of (S1) at prices $p = 0.845, 1.095, 1.400, 2.108$, and $x = 0.023, 0.092, 0.157, 0.266$, respectively. These points are chosen such that they correspond to the 24%, 48%, 72%, and 96% quantiles of the distribution of $p(T)$ and asymptotic distribution of $x(t)$, respectively. We find the following from these figures. First, at a given time and a given price level, the no transaction region becomes narrower when the spread gets larger, and the no transaction region moves from the negative to the positive when the spread turns from the negative to the positive. For example, at $t = 0.05$ and $p(t) = 0.845$, the no transaction region changes from $[-9.4, -8.0]$ at $x(t) = 0.023$ to $[-4.6, -3.4]$ at $x(t) = 0.092$, $[-0.7, 0.2]$ at $x(t) = 0.157$, and $[3.2, 3.7]$ at $x(t) = 0.266$. Second, at a given time and a given spread level, the no transaction region becomes narrower when the price $p(t)$ gets larger, and the no transaction region moves up when the price becomes larger. For instance, at $t = 0.05$ and $x(t) = 0.023$, the no transaction region changes from $[-9.4, -8.0]$ at $p(t) = 0.845$ to $[-6.8, -5.6]$ at $p(t) = 1.095$, $[-4.9, -3.9]$ at $p(t) = 1.400$, and $[-2.7, -2.0]$ at $p(t) = 2.108$. Note that the movement of the no transaction region with respect to price change but with a fixed spread level is relatively smaller than that with respect to spread change but with a fixed price level. Third, when time ellipses from 0 to 1, the no transaction region moves upward. For instance, at the fixed price-spread level $(p(t), x(t)) = (1.095, 0.092)$, the no transaction intervals at

$t = 0.05, 0.35, 0.65$ and $0.95$ are $[-2.6, -1.6]$, $[-2.1, -1.2]$, $[-1.5, -0.7]$, and $[-0.8, -0.2]$, respectively.

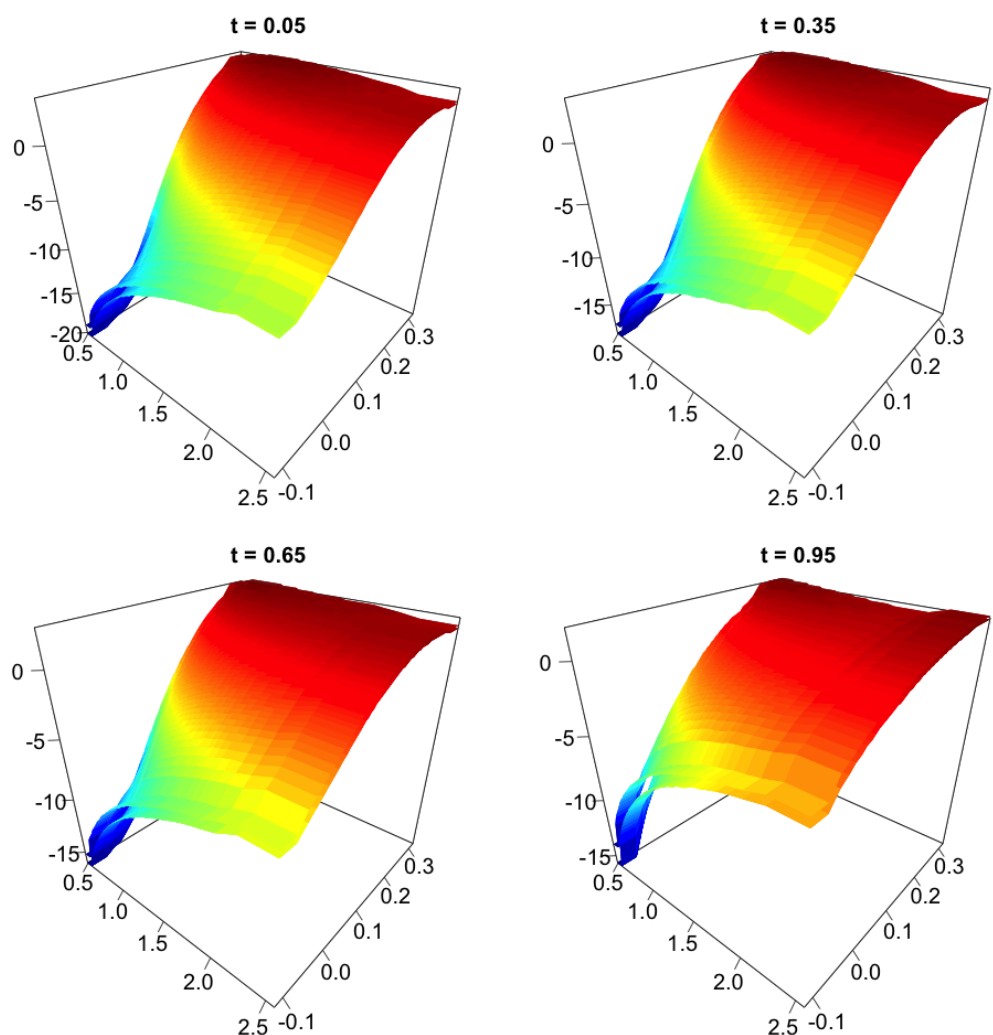

**Figure 1.** Buy and sell boundaries of the baseline scenario (S1) at different times.

We then discuss the impact of different parameter values on the buy and sell boundaries (or no transaction regions). Besides the parameter values in (S1), we now consider other 18 sets of parameter values, labeled as Scenarios 2–19. In each of Scenarios 2–19, all parameters values are same as those in (S1) except one parameter is changed as the specification; see Table 1 that summarizes parameter values in all 19 scenarios. For example, Scenario 2 uses parameter values $\mu = 0.1$ and assume all other parameters $\sigma, \theta, \kappa, \nu, \rho, r, \gamma$ and $\zeta_p = \zeta_q = \xi_p = \xi_q$ have same values as those in (S1). We discretize the state space $(t, p, x, y, g)$, and use the developed Markov chain approximation to solve the discretized optimization problem for Scenarios 2–19.

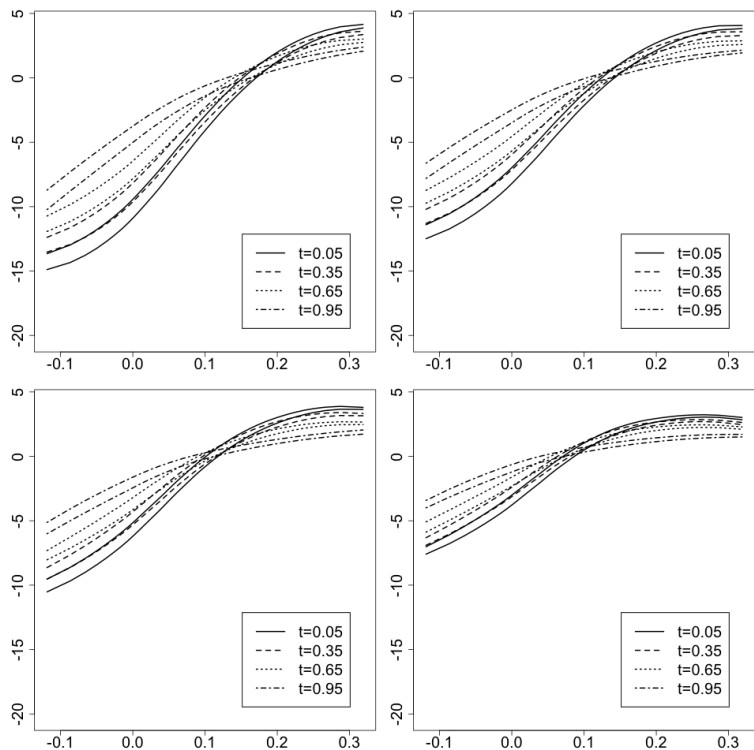

**Figure 2.** Buy and sell boundaries of at prices $P_t = 0.845$ (**top left**), 1.095 (**top right**), 1.400 (**bottom left**), and 2.108 (**bottom right**) and different times.

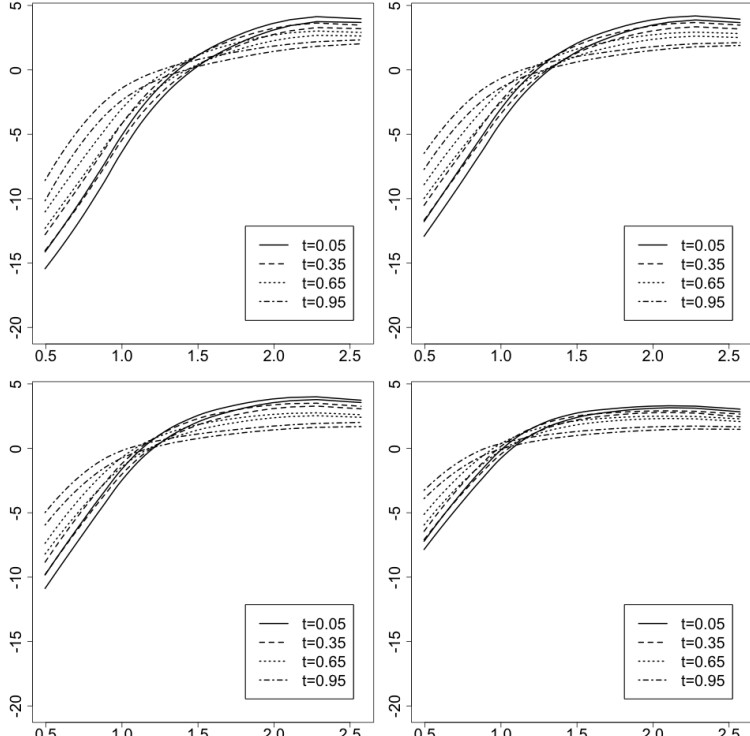

**Figure 3.** Buy and sell boundaries of at spread $X_t = 0.023$ (**top left**), 0.092 (**top right**), 0.157 (**bottom left**), and 0.266 (**bottom right**) and different times.

To compare the buy and sell boundaries (or no transaction regions) among different scenarios, we plot the buy and sell boundaries over time at four fixed points $(p^{(1)}, x^{(1)}) = (0.9, 0.09)$, $(p^{(2)}, x^{(2)}) = (0.9, 0.12)$, $(p^{(3)}, x^{(3)}) = (1.5, 0.09)$, and $(p^{(4)}, x^{(4)}) = (1.5, 0.12)$,

respectively. Figures 4–12 demonstrate variations of the buy and sell boundaries over time for different values of $\mu, \sigma, \theta, \kappa, \nu, \rho, r, \gamma, \zeta_p(= \zeta_q = \xi_p = \xi_q)$, respectively. In each figure, we plot the buy and sell boundaries for $(p^{(i)}, x^{(i)})$, $i = 1, 2, 3, 4$ on the top left, top right, bottom left, and bottom right, respectively, we also use the solid (dashed, dotted) lines to represent the baseline value (the smaller value, the larger value) of the parameter under comparison. Figure 4 suggests that when $\mu$ increases, the buy and sell boundaries move downward at all four points. Figure 5 indicates that when $\sigma$ increases, the buy and sell boundaries move upward at $(p^{(1)}, x^{(1)})$ and $(p^{(2)}, x^{(2)})$, but move downward at $(p^{(3)}, x^{(3)})$ and $(p^{(4)}, x^{(4)})$. Figure 6 shows that, when $\theta$ increases, the buy and sell boundaries move downward at all four points. Figure 7 indicates that, when $\kappa$ increases, the buy and sell boundaries move downward, and the magnitude of such movement is larger at $(p^{(1)}, x^{(1)})$ than the other three points. Figure 8 shows that, when $\nu$ increases, the buy and sell boundaries move upward at $(p^{(i)}, x^{(i)})$, $i = 1, 2, 3$, but move downward at $(p^{(4)}, x^{(4)})$. Figure 9 suggests that, when the correlation $\rho$ changes from the negative to the positive, the buy and sell boundaries move downwards at $(p^{(1)}, x^{(1)})$ and $(p^{(2)}, x^{(2)})$, but move upward at $(p^{(3)}, x^{(3)})$ and $(p^{(4)}, x^{(4)})$. Figure 10 indicates that variations of interest rate $r$ have little impact on the buy and sell boundaries. Figure 11 shows that, when the risk aversion parameter $\gamma$ increases, the buy and sell boundaries move upward at $(p^{(i)}, x^{(i)})$, $i = 1, 2, 3$, but move downward at $(p^{(4)}, x^{(4)})$. Figure 12 suggests that, when the transaction cost increases, the center of the no transaction region seems to not change, but the region gets wider.

**Table 1.** Parameter values of different scenarios.

| (S1) $\mu = 0.2, \sigma = 0.4, \theta = 0.1, \kappa = 1, \nu = 0.15, \rho = 0.5,$ | | |
| :--- | :--- | :--- |
| $r = 0.01, \gamma = 5$ and $\zeta_p = \zeta_q = \xi_p = \xi_q = 0.0005.$ | | |
| (S2) $\mu = 0.1$ | (S8) $\kappa = 0.8$ | (S14) $r = 0.005$ |
| (S3) $\mu = 0.3$ | (S9) $\kappa = 1.2$ | (S15) $r = 0.03$ |
| (S4) $\sigma = 0.2$ | (S10) $\nu = 0.1$ | (S16) $\gamma = 3$ |
| (S5) $\sigma = 0.6$ | (S11) $\nu = 0.2$ | (S17) $\gamma = 8$ |
| (S6) $\theta = -0.05$ | (S12) $\rho = -0.2$ | (S18) $\zeta_p = \zeta_q = \xi_p = \xi_q = 0.0001$ |
| (S7) $\theta = 0.3$ | (S13) $\rho = 0.6$ | (S19) $\zeta_p = \zeta_q = \xi_p = \xi_q = 0.0010$ |

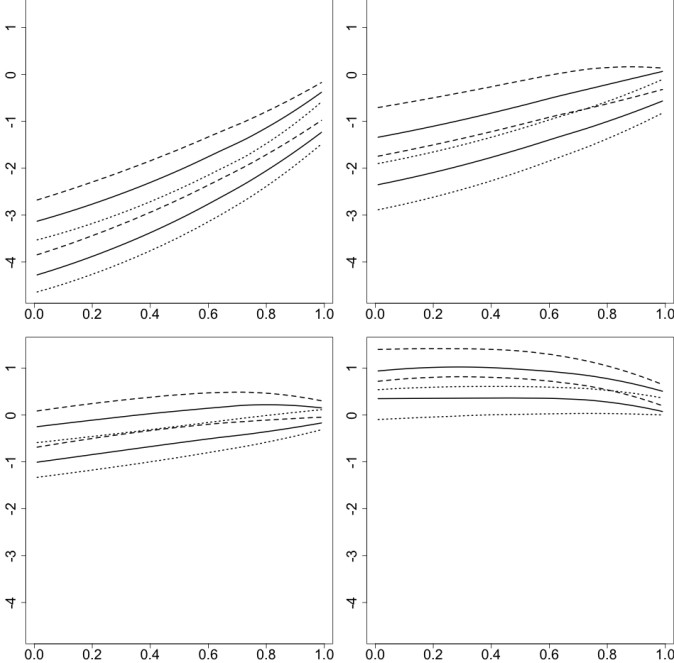

**Figure 4.** Buy and sell boundaries of at fixed prices $(p^{(i)}, x^{(i)})$, $i = 1, 2, 3, 4$ for $\mu = 0.1$ (dashed), 0.2 (solid), 0.3 (dotted).

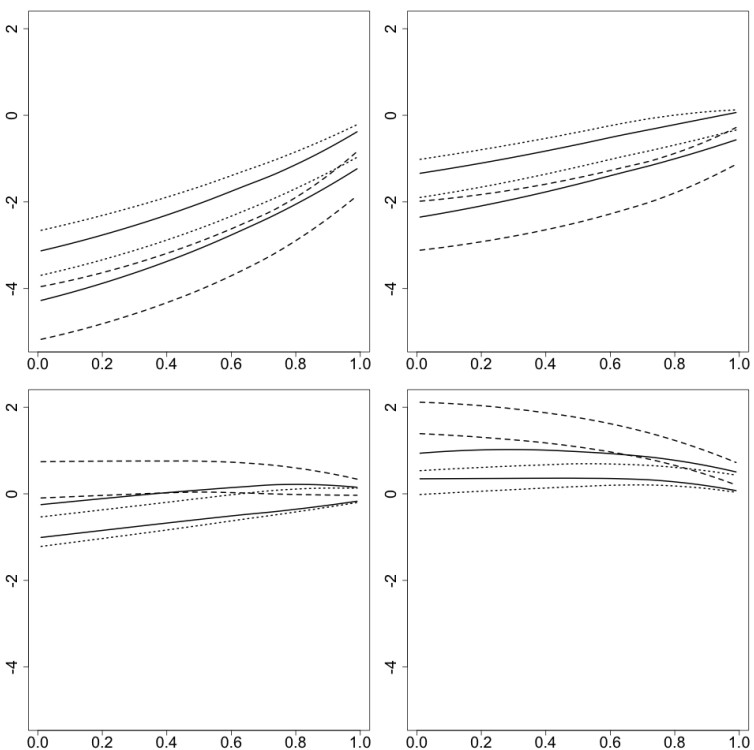

**Figure 5.** Buy and sell boundaries of at fixed price $(p^{(i)}, x^{(i)})$, $i = 1, 2, 3, 4$ for $\sigma = 0.2$ (dashed), 0.4 (solid), 0.6 (dotted).

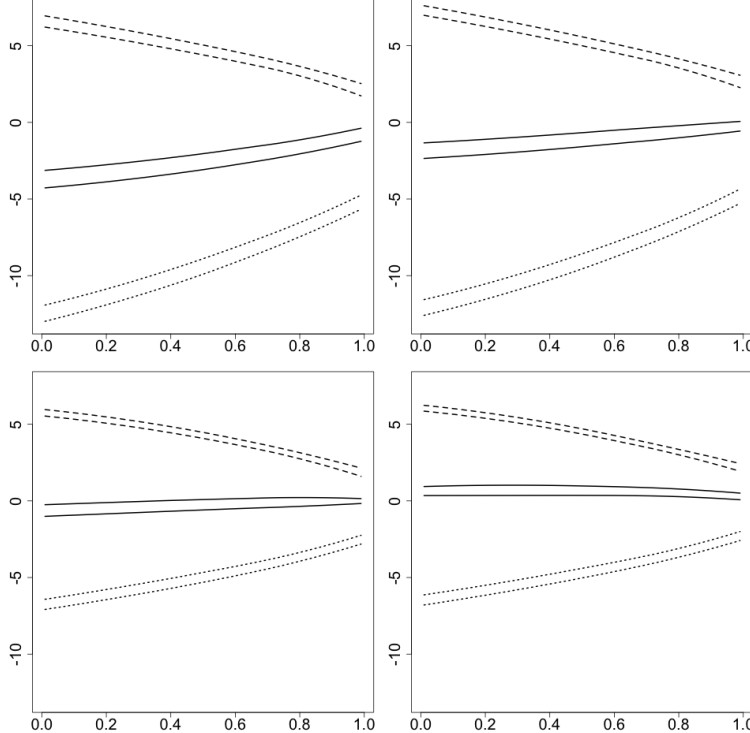

**Figure 6.** Buy and sell boundaries of at fixed price $(p^{(i)}, x^{(i)})$, $i = 1, 2, 3, 4$ for $\theta = -0.05$ (dashed), 0.1 (solid), 0.3 (dotted).

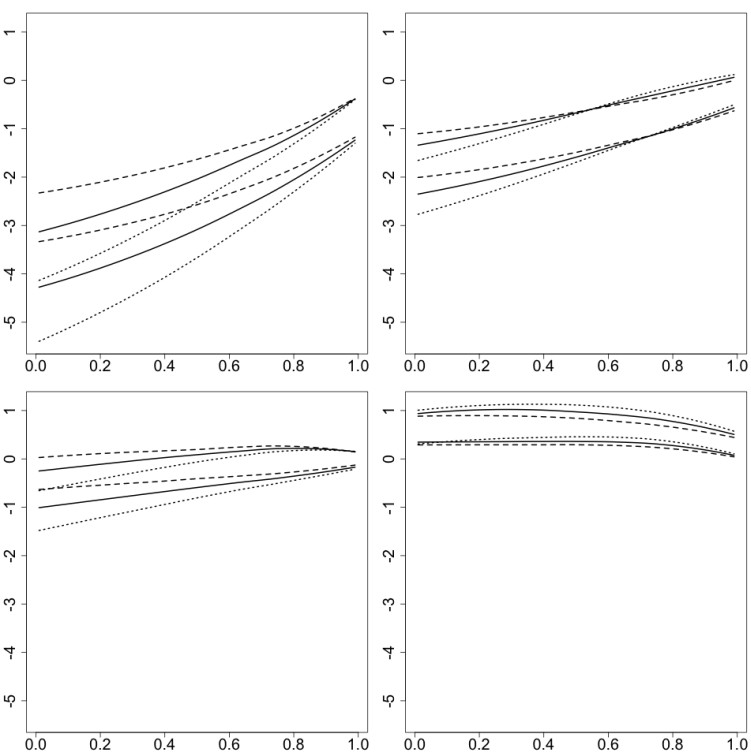

**Figure 7.** Buy and sell boundaries of at fixed price $(p^{(i)}, x^{(i)})$, $i = 1, 2, 3, 4$ for $\kappa = 0.8$ (dashed), 1 (solid), and 1.2 (dotted).

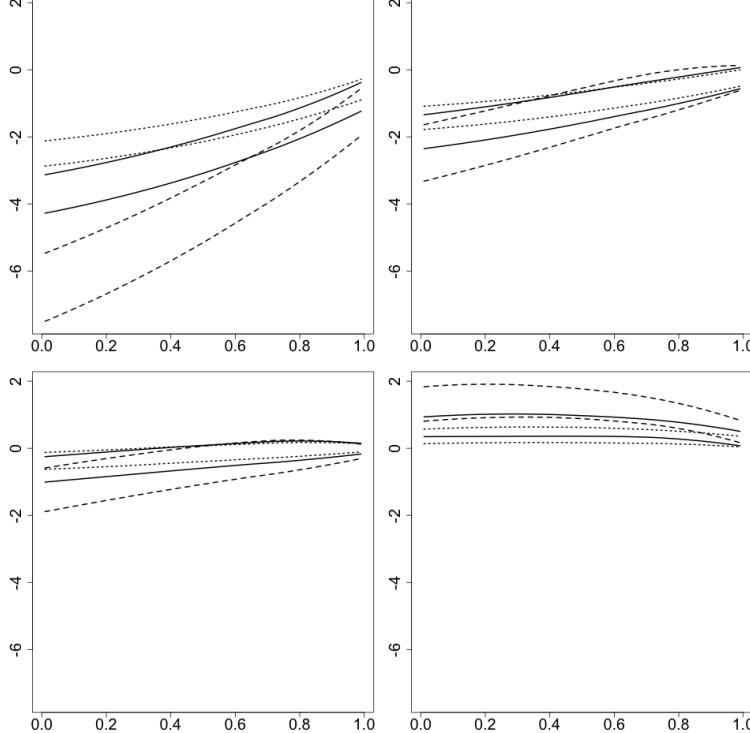

**Figure 8.** Buy and sell boundaries of at fixed price $(p^{(i)}, x^{(i)})$, $i = 1, 2, 3, 4$ for $\nu = 0.1$ (dashed), 0.15 (solid), and 0.2 (dotted).

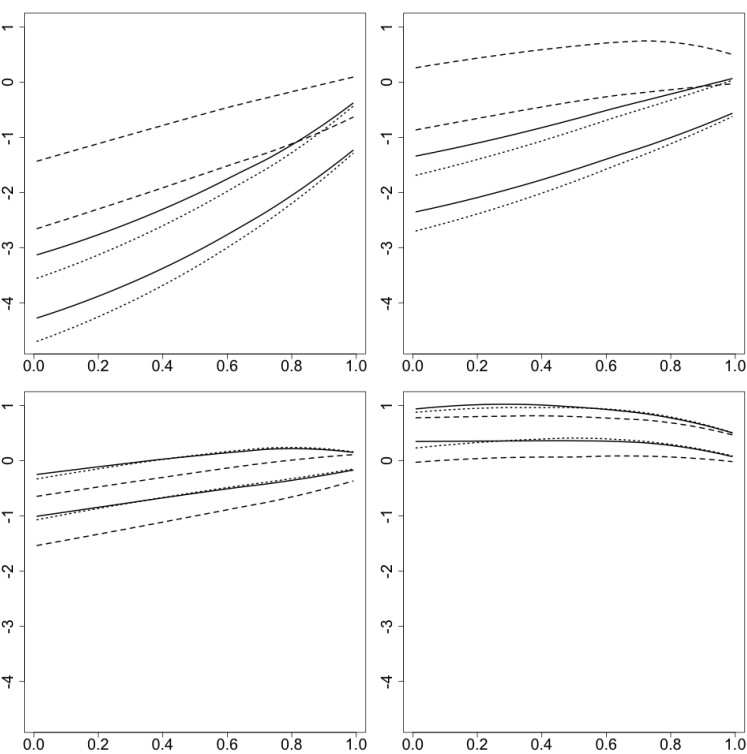

**Figure 9.** Buy and sell boundaries of at fixed price $(p^{(i)}, x^{(i)})$, $i = 1, 2, 3, 4$ for $\rho = -0.2$ (dashed), 0.5 (solid), and 0.6 (dotted).

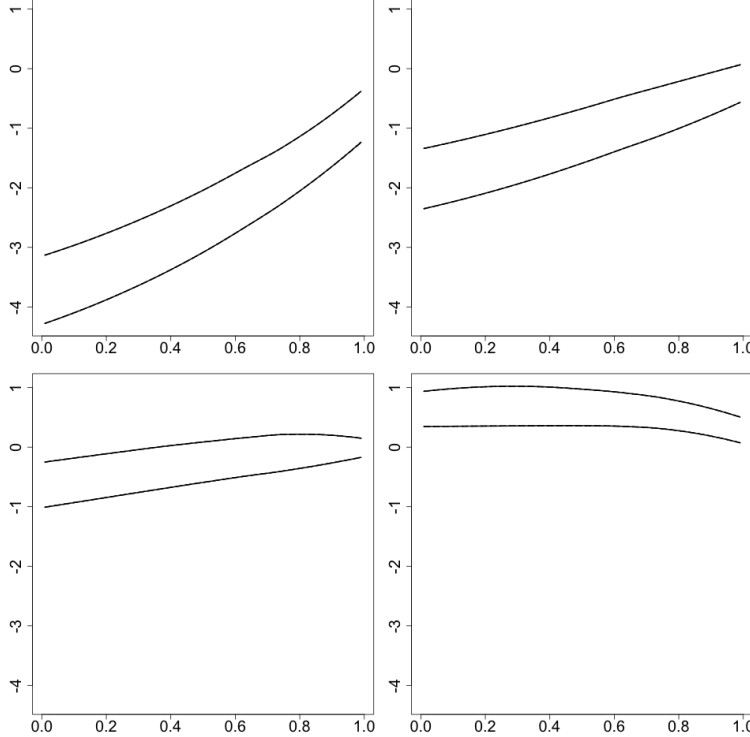

**Figure 10.** Buy and sell boundaries of at fixed price $(p^{(i)}, x^{(i)})$, $i = 1, 2, 3, 4$ for $r = 0.005$ (dashed), 0.01 (solid), and 0.03 (dotted).

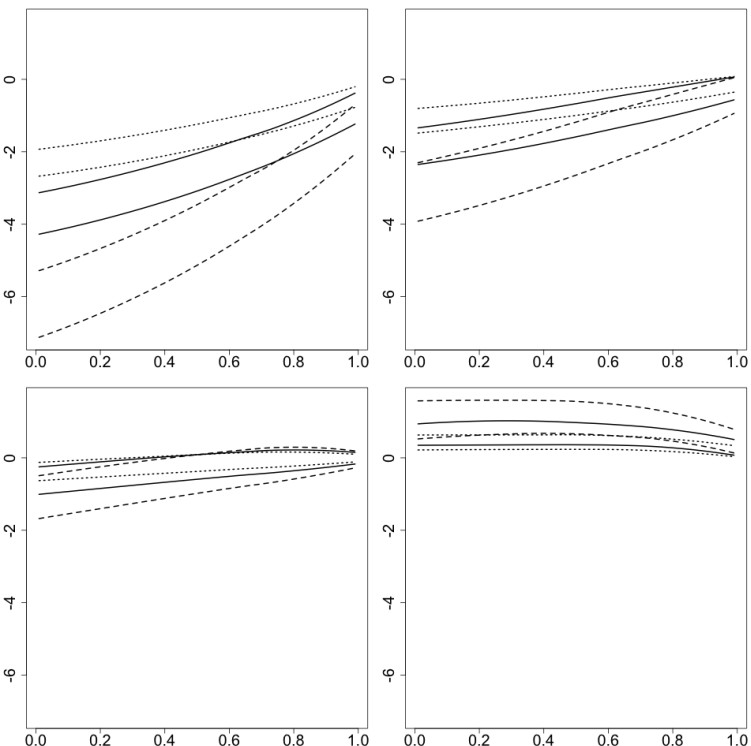

**Figure 11.** Buy and sell boundaries of at fixed price $(p^{(i)}, x^{(i)})$, $i = 1, 2, 3, 4$ for $\gamma = 3$ (dashed), 5 (solid), and 8 (dotted).

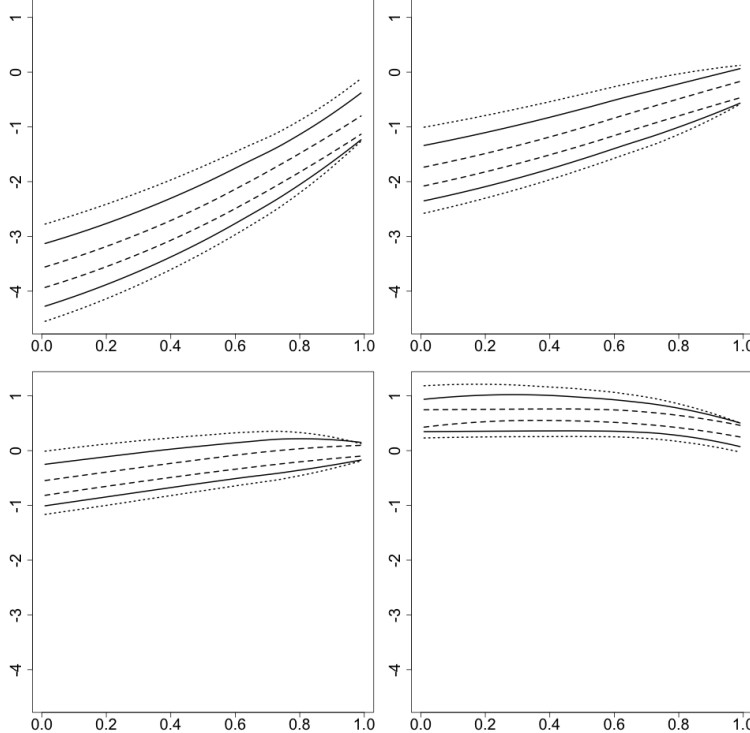

**Figure 12.** Buy and sell boundaries of at fixed price $(p^{(i)}, x^{(i)})$, $i = 1, 2, 3, 4$ for $\zeta_p = \zeta_q = \xi_p = \xi_q = 0.0001$ (dashed), 0.0005 (solid), and 0.0010 (dotted).

### 4.2. Performance of the Strategy

We also perform simulation studies to investigate the performance of the optimal trading strategy. For comparison purpose, we also consider a benchmark strategy that is

analogous to the relative-value arbitrage strategy used in Gatev et al. (2006) and based on standard deviation of the spread. Specifically, the strategy opens a position when the spread exceeds twice the standard deviation of the spread process, and closes the position when either price converges or the maturity is reached. As the benchmark strategy doesn't specify the number of shares of stocks that should be bought or sold, we assume that the number of shares of stocks traded each time is one.

We simulate the price process $p_t$ and the spread process $x_t$ to compare the performance of the benchmark strategy and our strategy in scenarios (S1)–(S19). Assume that $T = 1$, and we discretize the time interval $(0, 1]$ as $\{0.01, 0.02, \ldots, 0.99, 1\}$, so that we have 100 trading periods. For each scenario, we simulate 1000 paths of $\{(p_t, x_t)| t = 0, 0.01, \ldots, 0.99, 1, p_0 = 1\}$, and for each simulated path $(p_t, x_t)$, we implement the benchmark strategy and the optimal strategy at $t = 0.01, 0.02, \ldots, 0.99$ and close the position at $T = 1$. Let $i = b, o$ represent the benchmark and the optimal strategies, respectively. For each realized trading strategies, denote $N^{(i)}$ as the number of trades (i.e., buy and sell) among the 100 trading periods and $PL^{(i)} = -C^{(i)}(L_p, M_p; 0, 1)$ the total profit made during the trading process. Note that the benchmark strategy trades only one share of stock each time while the number of shares of stocks in the optimal strategy are "optimally" chosen based on the buy and sell regions, we define $PS^{(i)}$ as the the average profit (or loss) generated from the maximum number of shares of stocks during the trading process. That is, $PS^{(i)} := -C^{(i)}(L_p, M_p; 0, 1) / \max_t |Y_t^{(i)}|$, where $Y_t^{(i)}$ is the number of shares of stock $P$ at $t = 0.01, 0.02, \ldots, 0.99$.

Table 2 summarizes the mean and standard error of $N^{(i)}$, $PL^{(i)}$, and $PS^{(i)}$ ($i = o, b$) for 1000 paths in each scenario. We note that the total numbers of trades $N^{(o)}$ in the optimal strategy range from 45.736 to 55.821 for (S1)–(S17), and increases (or decreases) significantly when the transaction costs decreases (or increases) in (S18) and (S19). In comparison to this, the total numbers of trades $N^{(b)}$ in the benchmark strategy are much smaller, essentially, between 1 and 2. This suggests the benchmark strategy is much more conservative than the optimal strategy. For the realized profit over the trading period, $PL^{(o)}$ is much larger than $PL^{(b)}$ as the optimal strategy can choose to buy or sell the "optimal" number of shares of stock pairs, while the benchmark strategy only buy or sell one share of stock pair. $PS^{(o)}$ and $PS^{(b)}$ remove the impact of number of shares of traded stocks, and provide the average earning per traded stock, and we notice that $PS^{(o)}$ is still significantly higher than $PS^{(b)}$.

**Table 2.** Performance of strategies.

|  | $N^{(o)}$ | $PL^{(o)}$ | $PS^{(o)}$ | $N^{(b)}$ | $PL^{(b)}$ | $PS^{(b)}$ |
|---|---|---|---|---|---|---|
| (S1) | 52.289 (0.247) | 0.349 (0.019) | 0.048 (0.004) | 1.094 (0.084) | 0.005 (0.002) | 0.005 (0.002) |
| (S2) | 53.218 (0.241) | 0.389 (0.020) | 0.051 (0.004) | 1.094 (0.084) | 0.006 (0.002) | 0.006 (0.002) |
| (S3) | 51.348 (0.253) | 0.318 (0.019) | 0.046 (0.004) | 1.094 (0.084) | 0.004 (0.002) | 0.004 (0.002) |
| (S4) | 52.999 (0.208) | 0.378 (0.019) | 0.054 (0.003) | 1.094 (0.084) | 0.007 (0.002) | 0.007 (0.002) |
| (S5) | 51.896 (0.275) | 0.326 (0.019) | 0.040 (0.005) | 1.094 (0.084) | 0.003 (0.004) | 0.003 (0.004) |
| (S6) | 49.299 (0.235) | 0.357 (0.020) | 0.032 (0.003) | 1.094 (0.084) | 0.003 (0.002) | 0.003 (0.002) |
| (S7) | 54.233 (0.262) | 0.344 (0.019) | 0.064 (0.005) | 1.094 (0.084) | 0.008 (0.003) | 0.008 (0.003) |
| (S8) | 55.821 (0.304) | 0.359 (0.021) | 0.062 (0.006) | 1.094 (0.084) | 0.005 (0.003) | 0.005 (0.003) |
| (S9) | 45.736 (0.254) | 0.266 (0.016) | 0.046 (0.003) | 1.094 (0.084) | 0.005 (0.002) | 0.005 (0.002) |
| (S10) | 46.347 (0.292) | 0.228 (0.016) | 0.042 (0.005) | 1.052 (0.083) | 0.004 (0.002) | 0.004 (0.002) |
| (S11) | 57.689 (0.212) | 0.489 (0.022) | 0.053 (0.003) | 1.206 (0.084) | 0.007 (0.002) | 0.007 (0.002) |
| (S12) | 46.774 (0.248) | 0.325 (0.015) | 0.065 (0.003) | 1.140 (0.086) | 0.008 (0.001) | 0.008 (0.001) |
| (S13) | 53.516 (0.245) | 0.361 (0.020) | 0.045 (0.004) | 1.140 (0.087) | 0.006 (0.002) | 0.006 (0.002) |
| (S14) | 54.027 (0.232) | 0.579 (0.032) | 0.048 (0.004) | 1.094 (0.084) | 0.005 (0.002) | 0.005 (0.002) |
| (S15) | 50.031 (0.266) | 0.219 (0.012) | 0.049 (0.004) | 1.094 (0.084) | 0.005 (0.002) | 0.005 (0.002) |
| (S16) | 52.300 (0.247) | 0.347 (0.019) | 0.048 (0.004) | 1.094 (0.084) | 0.005 (0.002) | 0.005 (0.002) |
| (S17) | 52.261 (0.247) | 0.357 (0.019) | 0.050 (0.004) | 1.094 (0.084) | 0.006 (0.002) | 0.006 (0.002) |
| (S18) | 73.801 (0.286) | 0.339 (0.019) | 0.045 (0.004) | 1.094 (0.084) | 0.006 (0.002) | 0.006 (0.002) |
| (S19) | 42.996 (0.222) | 0.339 (0.019) | 0.049 (0.004) | 1.094 (0.084) | 0.004 (0.002) | 0.004 (0.002) |

## 5. Real Data Studies

We test our model with real market data in this section. We present the sample and explain our methodology first, and then show the results and discussion.

A key step of implementing pairs trading strategy is to select two stocks for pairs trading. Gatev et al. (2006) illustrate how this can be done by using stock price data. An alternative to this approach is to use fundamentals analysis to select two stocks that have almost the same risk factor exposures; see Vidyamurthy (2004). In this study, we consider a hybrid of these two approaches. Specifically, we restrict two stocks $P$ and $Q$ to belong to the same industry sector. Table 3 lists six pairs of stocks selected from four different sectors. For each pair of stocks $P$ and $Q$, we compute the spread by regressing log price of stock $Q$ on the log price of stock $P$, and the fitted values of the regression is considered as the "transformed" price of $P$. Figure 13 shows six pairs of the original prices of $Q$ and transformed prices of $P$ over time.

**Table 3.** Six pairs of stocks selected from different industries.

| Sector | Stock $Q$ | Stock $P$ |
|---|---|---|
| Consumer goods | Apple Inc. (AAPL) | Procter & Gamble Co. (PG) |
| Consumer goods | Coca-Cola Co. (KO) | PepsiCo, Inc. (PEP) |
| Technology | Alphabet Inc Class A (GOOGL) | Microsoft Corporation (MSFT) |
| Technology | AT&T Inc. (T) | Verizon Communications Inc. (VZ) |
| Industrial goods | Boeing Corporation (BA) | General Electric Company (GE) |
| Financial | Goldman Sachs Group Inc. (GS) | JPMorgan Chase & Co. (JPM) |

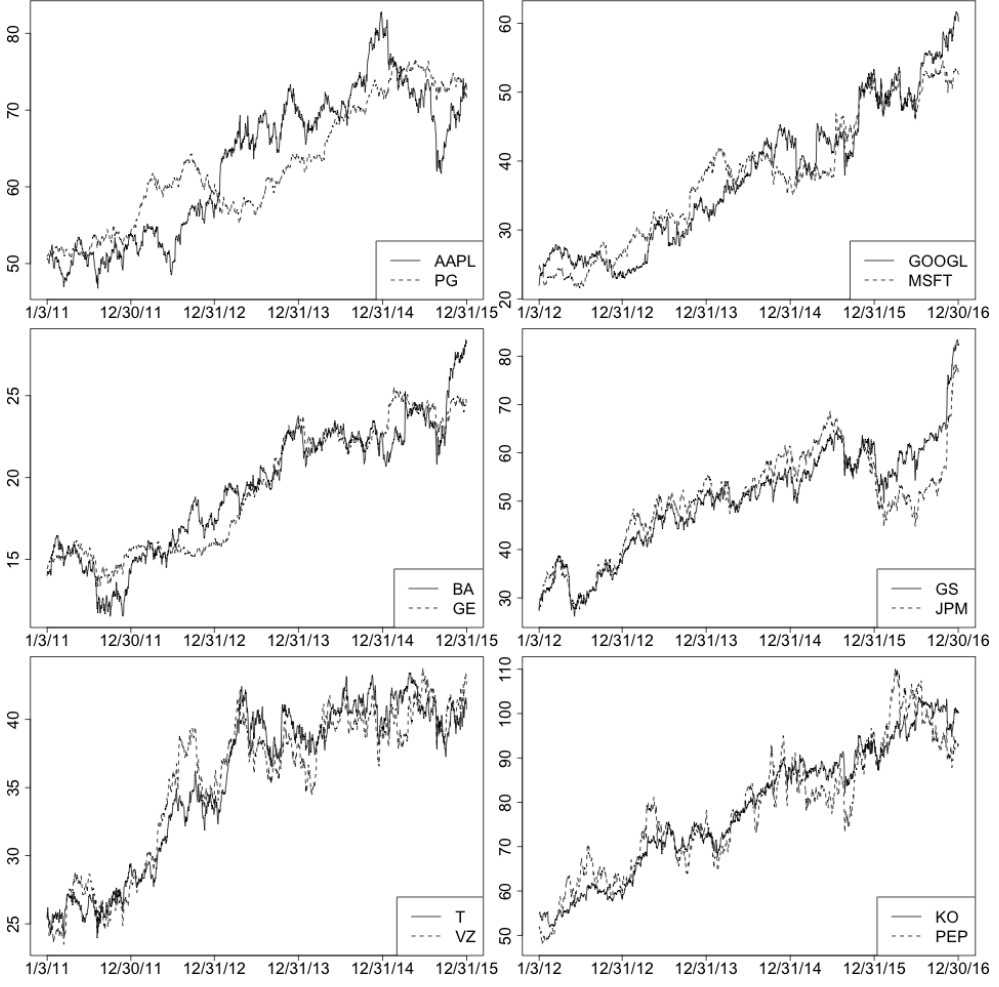

**Figure 13.** Original (solid) and transformed (dashed) prices of six pairs of stocks.

We then apply the optimal strategy and the benchmark strategy in Section 4.2 to test the out-of-the-sample performance. Specifically, we use the past three years of the historical data of each pair to estimate the model parameter, and run a unit-root test to conclude if the spread $x_t$ is a stationary process. If $x_t$ is not stationary, we do not implement any strategies. Otherwise, we implement both the optimal strategy and the benchmark strategy. Note that the optimal strategy can optimally choose the number of shares of stocks in each trade, while we still trade one unit of stock in the benchmark strategy. Table 4 shows the number of trades $N^{(i)}$, the accumulated profit (in U.S. dollars) at maturity $PL^{(i)}$, and the average profit per traded share $PS^{(i)}$ over two testing periods, for $i = o$ (the optimal strategy) and $i = b$ (the benchmark strategy). Table 4 suggests that the benchmark strategy is much more conservative than the optimal strategy. Besides, the average profits per traded share $PS^{(o)}$ of the optimal strategy are much larger than that of the benchmark strategy except for the stock pair $(KO, PEP)$.

**Table 4.** Performance of strategies.

| Pairs | Year | $N^{(o)}$ | $PL^{(o)}$ | $PS^{(o)}$ | $N^{(b)}$ | $PL^{(b)}$ | $PS^{(b)}$ |
|---|---|---|---|---|---|---|---|
| (AAPL, PG) | 2014 | 58 | 8.56 | 2.173 | 0 | 0 | 0 |
|  | 2015 | 70 | 25.439 | 3.91 | 0 | 0 | 0 |
| (BA, GE) | 2014 | 97 | 27.866 | 1.292 | 0 | 0 | 0 |
|  | 2015 | 165 | 168.543 | 1.982 | 20 | 0.455 | 0.455 |
| (T, VZ) | 2014 | 127 | 77.908 | 2.158 | 2 | 0.603 | 0.603 |
|  | 2015 | 131 | 115.587 | 2.883 | 0 | 0 | 0 |
| (GOOGL, MSFT) | 2015 | 103 | 94.271 | 6.734 | 8 | 1.623 | 1.623 |
|  | 2016 | 135 | 65.957 | 6.296 | 0 | 0 | 0 |
| (GS, JPM) | 2015 | 100 | 7.654 | 0.195 | 6 | −2.54 | −2.54 |
|  | 2016 | 200 | 94.542 | 2.375 | 8 | −1.66 | −1.66 |
| (KO, PEP) | 2015 | 142 | 37.51 | 0.675 | 22 | 10.154 | 10.154 |
|  | 2016 | 165 | 217.878 | 4.059 | 4 | 5.983 | 5.983 |

## 6. Concluding Remark

The problem of optimal pairs trading has been studied by many academic researchers and financial practitioners. Existing models and methods try to find either the optimal shares of stocks by assuming no transaction costs, or the optimal timing of trading fixed number of shares of stocks with transaction costs. In contrast to these analysis, this paper studies the joint effect of optimal shares and optimal trading times in pairs trading process with proportional transaction costs. Under the assumption that the investor's aim is to maximize the expected utility of terminal wealth, the optimal pair trading problem can be written as a singular stochastic control problem and solved by the approach in Davis et al. (1993). We then demonstrate the advantage of joint consideration of optimal shares and optimal trading times in pair trading via simulation and empirical studies.

The following issues may need further investigation to make this study more practical. First, our approach can be easily extended for nonexponential utility functions. In such a case, the optimization problem involves five (instead of four) variables, and the numerical algorithm in our paper needs to be modified to adapt for five variables. Second, our approach can be extended to solve the optimal co-integration trading, which involves $n$ stocks with $m$ co-integration relationship. Third, many empirical studies suggest that stock price processes can be better approximated by incorporating jumps. Using the framework and algorithms developed in Xing et al. (2017), the method developed here can be extended to the case that price processes follow geometric jump-diffusion processes. In such a case, the value function of the corresponding variational inequalities involve integro-differential equations, which can be solved by extending our numerical algorithm.

**Funding:** This research received no external funding.

**Institutional Review Board Statement:** Not applicable.

**Informed Consent Statement:** Not applicable.

**Data Availability Statement:** Data available on request.

**Conflicts of Interest:** The author declares no conflict of interest.

## Appendix A. Proof of Theorems

**Proof of Theorem 1.** In our case, the state $\mathbf{X}$ is $(s, \mathbf{x})$, where $\mathbf{x} = (p, x, y, g)$. Let $\mathbf{X}_0 = (s_0, p_0, x_0, y_0, G_0)$, it follows that there exists an optimal trading strategy, dictated by the pair of processes $(L_p^*(t), M_p^*(t))$, where $\mathbf{X}_0^*(t) = (t, p_0^*(t), x_0^*(t), y_0^*(t), g_0^*(t))$ is the optimal trajectory, with $\mathbf{X}_0^*(s_0) = \mathbf{X}_0$.

(i) First, we prove that $V$ is a viscosity subsolution of (21) on $[0, T] \times \mathbb{R}^+ \times \mathbb{R} \times \mathbb{R} \times \mathbb{R}$. For this, we must show that, for all smooth functions $\phi(\mathbf{X})$, such that $V(\mathbf{X}) - \phi(\mathbf{X})$ has a local maximum at $\mathbf{X}_0$, the following inequality holds:

$$\min\left\{-\mathcal{B}\phi(\mathbf{X}_0), \mathcal{S}\phi(\mathbf{X}_0), -\mathcal{L}\phi(\mathbf{X}_0)\right\} \leq 0. \tag{A1}$$

Without loss of generality, we assume that $V(\mathbf{X}_0) = \phi(\mathbf{X}_0)$ and $V \leq \phi$ on $[0, T] \times \mathbb{R}^+ \times \mathbb{R} \times \mathbb{R} \times \mathbb{R}$. We argue by contradiction: if the arguments inside the operator of (A1) satisfy $-\mathcal{B}\phi(\mathbf{X}_0) > 0$ and $\mathcal{S}\phi(\mathbf{X}_0) > 0$, then there exists $\theta > 0$, such that $-\mathcal{L}\phi(\mathbf{X}_0) > \theta$. From the fact that $\phi$ is smooth, the above inequalities become $-\mathcal{B}\phi(\mathbf{X}) > 0$, $\mathcal{S}\phi(\mathbf{X}) > 0$, and $-\mathcal{L}\phi(\mathbf{X}) > \theta$, where $\mathbf{X} = (t, p, x, y, g) \in \mathscr{B}(\mathbf{X}_0)$, a neighborhood of $\mathbf{X}_0$. In Lemma 1, it is shown that $\mathbf{X}_0^*(t)$ has no jumps, P-a.s., at $\mathbf{X}_0 = \mathbf{X}_0^*(s_0)$. Hence, $\tau(\omega)$, defined by

$$\tau(\omega) = \inf\{t \in (s_0, T] : \mathbf{X}_0^*(t) \notin \mathscr{B}(\mathbf{X}_0)\},$$

is positive P-a.s., and therefore the integral along $\mathbf{X}_0^*(t)$

$$-\theta \int_{s_0}^{\tau} dt > E \int_{s_0}^{\tau} \mathcal{B}\phi(\mathbf{X}_0^*(t)) dL^*(t) - E \int_{s_0}^{\tau} \mathcal{S}\phi(\mathbf{X}_0^*(t)) dM^*(t) + E \int_{s_0}^{\tau} \mathcal{L}\phi(\mathbf{X}_0^*(t)) dt$$
$$= E\{I_1\} - E\{I_2\} + E\{I_3\}, \tag{A2}$$

where $(L^*(t), M^*(t))$ is the optimal trading strategy at $\mathbf{X}_0$. Applying Itô's formula to $\phi(\mathbf{X})$, where the state dynamics are given by (1)–(6), we get

$$E\{\phi(\mathbf{X}_0^*(\tau))\} = \phi(\mathbf{X}_0) + E\{I_1\} - E\{I_2\} + E\{I_3\}. \tag{A3}$$

Since $V(\mathbf{X}) \leq \phi(\mathbf{X})$, for all $\mathbf{X} \in \mathcal{B}(\mathbf{X}_0)$, and $V(\mathbf{X}_0) = \phi(\mathbf{X}_0)$, (A2) and (A3) yield

$$E\{V(\mathbf{X}_0^*(\tau))\} \leq V(\mathbf{X}_0) + E\{I_1\} - E\{I_2\} + E\{I_3\} < V(\mathbf{X}_0) - \theta \int_{s_0}^{\tau} dt,$$

which violates the dynamic programming principle, together with the optimality of $(L^*(t), M^*(t))$. Therefore, at least one of the arguments inside the minimum operator of (A1) is nonpositive, and hence the value function is a viscosity subsolution of (21).

(ii) In the second part of the proof, we show that $V$ is a viscosity supersolution of (21). For this, we must show that, for all smooth functions $\phi(\mathbf{X})$, such that $V(\mathbf{X}) - \phi(\mathbf{X})$ has a local minimum at $\mathbf{X}_0$, the following inequality holds:

$$\min\left\{-\mathcal{B}\phi(\mathbf{X}_0), \mathcal{S}\phi(\mathbf{X}_0), -\mathcal{L}\phi(\mathbf{X}_0)\right\} \geq 0, \tag{A4}$$

where, without loss of generality, $V(\mathbf{X}_0) = \phi(\mathbf{X}_0)$ and $V(\mathbf{X}) \geq \phi(\mathbf{X})$ on $[0, T] \times \mathbb{R}^+ \times \mathbb{R} \times \mathbb{R} \times \mathbb{R}$. In this case, we prove that each argument of the minimum operator of (A4) is non-negative.

Consider the trading strategy $L(t) = L_0 > 0$, $s_0 \leq t \leq T$, and $M(t) = 0$, $s_0 \leq t \leq T$. By the dynamic programming principle,

$$V(s_0, p_0, x_0, y_0, g_0) \geq V(s_0, p_0, x_0, y_0 + L_0, g - (a_p - b_q e^{X_0})p_0 L_0).$$

This inequality holds for $\phi(s, p, x, y, g)$ as well, and, by taking the left-hand side to the right-hand side, dividing by $L_0$, and sending $L_0 \to 0$, we get $\mathcal{B}\phi(\mathbf{X}_0) \leq 0$. Similarly, by using the trading strategy $L(t) = 0, s_0 \leq t \leq T$, and $M(t) = M_0 > 0, s_0 \leq t \leq T$, the second argument inside the minimum operator is found to be non-negative.

Finally, consider the case where no trading is applied. By the dynamic programming principle

$$E\{V(\mathbf{X}_0^d(t))\} \leq V(s_0, p_0, x_0, y_0, g_0), \tag{A5}$$

where $\mathbf{X}_0^d(t)$ is the state trajectory of starting at $s_0$, when $M(t) = L(t) = 0$, $s_0 \leq t \leq T$, given by (1)–(6) as

$$\mathbf{X}_0^d(t) = (t, p(t), x(t), y_0, g(t))$$

and $\mathbf{X}_0^d(t) \in \mathscr{B}(\mathbf{X}_0)$. Therefore, by applying Itô's rule on $\phi(s, X, B, y, G)$, inequality (A5) yields

$$E\left\{\int_{s_0}^t \mathcal{L}\phi(\mathbf{X}_0^d(\xi))d\xi\right\} \leq 0,$$

and, by letting $t \to s_0$, the third argument inside the minimum operator is found to be non-negative. This complete the proof. $\quad\square$

**Lemma A1.** *Assume that $-\mathcal{B}\phi(\mathbf{X}_0) > 0$, and denote the event that the optimal trajectory $\mathbf{X}_0^*(t)$ has a jump of size $\epsilon$, along the direction $(0, 0, 0, 1, -(a_p - b_q e^{x_0})p_0)$ by $A(\omega)$. Assume that the state (after the jump) is $(s_0, p_0, x_0, y_0 + \epsilon, -(a_p - b_q e^{x_0})B_0\epsilon) \in \mathscr{B}(\mathbf{X}_0)$. Then,*

$$\left(\mathcal{B}\phi(\mathbf{X}_0)\right)P(A) \geq 0, \tag{A6}$$

*therefore $P(A) = 0$. Similarly, if $\mathcal{S}\phi(\mathbf{X}_0) > 0$, then the optimal trajectory has no jumps along the direction $(0, 0, 0, -1, (b_p - a_q e^{x_0})p_0)$, P-a.s. at $\mathbf{x}_0$.*

**Proof.** By the principle of dynamic programming,

$$V(s_0, p_0, x_0, y_0, g_0) = E\{V(s_0, p_0, x_0, y_0 + \epsilon, -(a_p - b_q e^{x_0})B_0\epsilon)\}$$
$$= \int_{A(\omega)} V(s_0, p_0, x_0, y_0 + \epsilon, -(a_p - b_q e^{x_0})B_0\epsilon)dP + \int_{A(\omega)} V(s_0, p_0, x_0, y_0, g_0)dP,$$

and therefore

$$\int_{A(\omega)} \left[\phi(s_0, p_0, x_0, y_0 + \epsilon, -(a_p - b_q e^{x_0})B_0\epsilon) - \phi(s_0, p_0, x_0, y_0, g_0)\right]dP \geq 0,$$

since $V(\mathbf{X}) \leq \phi(\mathbf{X})$ for all $\mathbf{X} \in \mathcal{B}(\mathbf{X}_0)$ and $V(\mathbf{X}_0) = \phi(\mathbf{X}_0)$. Therefore,

$$\limsup_{\epsilon \to 0} \left\{\int_{A(\omega)} \frac{\phi(s_0, p_0, x_0, y_0 + \epsilon, -(a_p - b_q e^{x_0})p_0\epsilon) - \phi(s_0, p_0, x_0, y_0, g_0)}{\epsilon}dP\right\} \geq 0,$$

and, by Fatou's lemma,

$$\int_{A(\omega)} \limsup_{\epsilon \to 0} \left\{\frac{\phi(s_0, p_0, x_0, y_0 + \epsilon, -(a_p - b_q e^{x_0})B_0\epsilon) - \phi(s_0, p_0, x_0, y_0, G_0)}{\epsilon}\right\}dP \geq 0,$$

which implies (A6).  □

**Proof of Theorem 3.**  Let

$$V^\delta(t, p, x, y, g) = \begin{cases} \mathbb{V}^\delta(\chi, \mathbb{p}, \mathbb{x}, y, g) & \text{if } t \in [\chi, \chi + \delta), y \in [\nu, \nu + \kappa\delta), \\ Z(\mathbb{p}, \mathbb{x}, y, g) & \text{if } t = T \end{cases}$$

and

$$\underline{V}(\mathbf{X}) = \lim_{\substack{\mathbf{Y} \to \mathbf{X} \\ \delta \to 0}} \inf \{\mathbb{V}^\delta(\mathbf{Y})\} \quad \text{and} \quad \overline{V}(\mathbf{X}) = \lim_{\substack{\mathbf{Y} \to \mathbf{X} \\ \delta \to 0}} \sup \{\mathbb{V}^\delta(\mathbf{Y})\}, \tag{A7}$$

where $\mathbf{X} = (t, p, x, y, g)$. We will show that $\underline{V}(\mathbf{X})$ and $\overline{V}(\mathbf{X})$ are a viscosity supersolution and a viscosity subsolution of (21), respectively. Combining this with the uniqueness of the viscosity solution of (21) yields $\underline{V}(\mathbf{X}) \geq \overline{V}(\mathbf{X})$ on $[0, T] \times \mathbb{R}^+ \times \mathbb{R} \times \mathbb{R} \times \mathbb{R}$. The opposite inequality is true by the definition of $\underline{V}(\mathbf{X})$ and $\overline{V}(\mathbf{X})$, and therefore

$$\underline{V}(\mathbf{X}) = \overline{V}(\mathbf{X}) = V(\mathbf{X}),$$

which, together with (A7), also implies the local uniform convergence of $\mathbb{V}^\delta$ to $V$.

Note that we only prove that $\underline{V}$ is a viscosity supersolution of (21), as the arguments for $\overline{V}$ is identical. Let $\mathbf{X}_0$ be a local minimum of $\underline{V} - \phi$ on $[0, T] \times \mathbb{R}^+ \times \mathbb{R} \times \mathbb{R} \times \mathbb{R}$, for $\phi \in C^{1,2}([0, T] \times \mathbb{R}^+ \times \mathbb{R} \times \mathbb{R} \times \mathbb{R})$. Without loss of generality, we may assume that $\mathbf{X}_0$ is a strict local minimum, that $\underline{V}(\mathbf{X}_0) = \phi(\mathbf{X}_0)$, and that $\phi \leq -2 \times \sup_\delta \{||\mathbb{V}^\delta||_\infty\}$ outside the vall $\mathscr{B}(\mathbf{X}_0, R)$, $R > 0$, where $\underline{V}(\mathbf{X}) - \phi(\mathbf{X}) \geq 0$.

Then, there exist sequences $\delta_n \in \mathbb{R}^+$ and $\mathbf{Y}_n \in [0, T] \times \mathbb{R}^+ \times \mathbb{R} \times \mathbb{R} \times \mathbb{R}$, such that

$$\delta_n \to 0, \mathbf{Y}_n \to \mathbf{X}_0, \mathbb{V}^{\delta_n}(\mathbf{Y}_n) \to \underline{V}(\mathbf{X}_0), \mathbf{Y}_n \text{ if a global minimum point of } \mathbb{V}_j^{\delta_n} - \phi.$$

Let $h_n = \mathbb{V}^{\delta_n} - \phi$; then

$$h_n \to 0 \text{ and } \mathbb{V}_j^{\delta_n}(\mathbf{X}) \geq \phi(\mathbf{X}) + h_n(\mathbf{X}) \quad \text{for any } \mathbf{X} \in [0, T] \times \mathbb{R}^+ \times \mathbb{R} \times \mathbb{R} \times \mathbb{R}. \tag{A8}$$

To show that $\underline{V}$ is a viscosity supersolution of (21), it suffices to show that

$$\min \left\{ -\mathcal{B}\phi(\mathbf{X}_0), \mathcal{S}\phi(\mathbf{X}_0), -\mathcal{L}\phi(\mathbf{X}_0) \right\} \geq 0. \tag{A9}$$

Let $\mathbb{Y}_n = (s_i, \mathbb{p}_n, \mathbb{x}_n, y_n, g_n)$, where $s_i \in [\chi_i, \chi_i + \delta_n)$ and $y_{\delta_n} \in [\vartheta_n, \vartheta_n + \kappa\delta_n)$. Denote $\mathbb{Y}_n^{(0)} = (\chi_n, \mathbb{p}_n, \mathbb{x}_n, y_n, g_n)$,

$$\mathbb{Y}_n^{(1)} = \left(\chi_n, \mathbb{p}_n, \mathbb{x}_n, \vartheta_n + \kappa\delta_n, g_n - (a_p - b_q e^{\mathbb{x}_n})\mathbb{p}_n\kappa\delta_n\right),$$

$$\mathbb{Y}_n^{(2)} = \left(\chi_n, \mathbb{p}_n, \mathbb{x}_n, \vartheta_n - \kappa\delta_n, g_n + (b_p - a_q e^{\mathbb{x}_n})\mathbb{p}_n\kappa\delta_n\right).$$

Then,

$$\mathbb{V}^{\delta_n}(\mathbb{Y}_n^{(0)}) = \max \left\{ \mathbb{V}^{\delta_n}(\mathbb{Y}_n^{(1)}), \mathbb{V}^{\delta_n}(\mathbb{Y}_n^{(2)}), E\{\mathbb{V}^{\delta_n}(\mathbb{Y}_{n+1}^{(0)})\} \right\}.$$

Now, we look at the following three cases.

*Case 1.* It holds that $\mathbb{V}^{\delta_n}(\mathbb{Y}_n^{(0)}) = \mathbb{V}^{\delta_n}(\mathbb{Y}_n^{(1)})$. Then (A8) implies that

$$\mathbb{V}^{\delta_n}(\mathbb{Y}_n^{(0)}) \geq \phi(\mathbb{Y}_n^{(1)}) + \mathbb{V}^{\delta_n}(\mathbb{Y}_n^{(0)}) - \phi(\mathbb{Y}_n^{(0)}),$$

and therefore

$$0 \geq \liminf_n \left\{ \frac{\phi(\mathbb{Y}_n^{(1)}) - \phi(\mathbb{Y}_n^{(0)})}{\delta_n} \right\} \geq \lim_{\delta \to 0} \inf \left\{ \frac{\phi(\mathbb{Y}_0^{(1)}) - \phi(\mathbb{Y}_0^{(0)})}{\delta} \right\}$$
$$= \frac{\partial \phi(\mathbf{X}_0)}{\partial y} - (a_p - e^{x_0(t)}) p_0(t) \frac{\partial \phi(\mathbf{x}_0)}{\partial g}.$$

*Case 2.* It holds that $\mathbb{V}^{\delta_n}(\mathbb{Y}_n^{(0)}) = \mathbb{V}^{\delta_n}(\mathbb{Y}_n^{(2)})$. Arguing similarly to case 1, we get

$$0 \geq -\left( \frac{\partial \phi(\mathbf{X}_0)}{\partial y} - (b_p - a_q e^{x_0(t)}) p_0(t) \frac{\partial \phi(\mathbf{X}_0)}{\partial g} \right).$$

*Case 3.* It holds that $\mathbb{V}^{\delta_n}(\mathbb{Y}_n^{(0)}) = E\{ \mathbb{V}^{\delta_n}(\mathbb{Y}_{n+1}^{(0)}) \}$. Then (A8) implies that

$$\mathbb{V}^{\delta_n}(\mathbb{Y}_n^{(0)}) \geq E\{ \phi(\mathbb{Y}_{n+1}^{(0)}) \} + \mathbb{V}^{\delta_n}(\mathbb{Y}_n^{(0)}) - \phi(\mathbb{Y}_{n+1}^{(0)}),$$

and therefore

$$0 \geq \liminf_n \left\{ \frac{\phi(\mathbb{Y}_{n+1}^{(0)}) - \phi(\mathbb{Y}_n^{(0)})}{\delta_n} \right\} \geq \lim_{\delta \to 0} \inf \left\{ \frac{\phi(\mathbb{Y}_1^{(0)}) - \phi(\mathbb{Y}_0^{(0)})}{\delta} \right\} = \mathcal{L}\phi(\mathbf{X}_0).$$

Combining the results in cases 1–3 yields (A9), and the proof is complete. □

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
