# Peer review of "A Singular Stochastic Control Approach for Optimal Pairs Trading with Proportional Transaction Costs"

_jrfm, doi:10.3390/jrfm15040147_

Round 1
Reviewer 1 Report
In this study, it is necessary to match the literature study by suggesting which areas of optimal trading strategies for pairs trading need any developments. The author mentions that this problem is widely investigated but then mentions just several publications. The lack of study in the field and the novelty of the author’s approach is not obvious.
Further explanation is needed to understand the target audience for this study. Does this approach have any practical potential? Or it is just another theoretical article with some ideas that will never work in trading?
For now, we can see some good math here, but what is the novelty? What can be done with results of this calculations?
The Introduction, Literature Review and Conclusion should be reconsidered by the author in order to be published.
Author Response
Please see the attached file which includes my point-to-point response and revision.

Reviewer 2 Report
An investor needs a rule to determine the number of shares of stocks P and
Q bought or sold at time instant t.
The investor's goal is to maximize the terminal wealth.
The problem is transformed into a singular stochastic control problem.
The author shows that the solution of the control problem is the
viscosity solution of a variational inequality equivalent to a
free boundary problem.
A numerical scheme is proposed based on
the discrete time dynamic programming principle.
A convergence result is proved.
Some buy and sell boundaries are computed and presented for 19 scenarios.
The performances of the optimal strategy is tested for six pairs of U.S. stocks.
Author Response
Thanks for your response. I have corrected the typos in the paper and make some changes to highlight the contribution of the paper. (The red part is the revision, the blue part marks the deletion from the previous version).

Round 2
Reviewer 1 Report
The author's position looks more clear now. As i have said in first round, the math is good, and it stands unchanged. I still think that the article can be improved in order to make it more easy to read, but I do not see the reason to reject it. And of course even now the article can be interesting for highly specialized and professional audience. It deserves to be published.